# Antibody Responses Following Primary Immunization with the Recombinant Herpes Zoster Vaccine (Shingrix^®^) in VZV Seronegative Immunocompromised Adults

**DOI:** 10.3390/vaccines13070737

**Published:** 2025-07-08

**Authors:** Andrea Wessely, Ines Zwazl, Melita Poturica, Lukas Weseslindtner, Michael Kundi, Ursula Wiedermann, Angelika Wagner

**Affiliations:** 1Center for Pathophysiology, Infectiology and Immunology, Institute of Specific Prophylaxis and Tropical Medicine, Medical University of Vienna, 1090 Wien, Austria; andrea.wessely@meduniwien.ac.at (A.W.); ines.zwazl@meduniwien.ac.at (I.Z.); melita.poturica@meduniwien.ac.at (M.P.); ursula.wiedermann@meduniwien.ac.at (U.W.); angelika.wagner@meduniwien.ac.at (A.W.); 2Center for Virology, Medical University of Vienna, 1090 Wien, Austria; lukas.weseslindtner@meduniwien.ac.at; 3Center for Public Health, Medical University of Vienna, 1090 Wien, Austria

**Keywords:** immunosuppression, hematopoietic stem cell transplantation (HSCT), VZV seronegative, VZV subunit vaccine, adjuvanted, seroconversion

## Abstract

**Background:** Immunocompromised patients are at risk of severe varicella zoster virus (VZV) infection and reactivation. In VZV seronegative immunocompromised persons, live-attenuated VZV vaccination is contraindicated, thus the recombinant herpes zoster vaccine (rHZV) remains a safe alternative, although an off-label application. Yet, data on the induction of a VZV-specific immune response in immunocompromised individuals with VZV-specific IgG below the assay’s cut-off are only available for patients after solid-organ transplantation (SOT). **Methods**: We retrospectively analyzed the induction of VZV-specific IgG antibody levels after vaccination with rHZV in immunocompromised patients who previously tested anti-VZV-IgG negative between March 2018 and January 2024. **Results**: Of 952 vaccinees screened that received 2 or 3 doses rHZV, depending on the underlying disease, 33 patients (median age 53.0; 51.5% female) with either hematopoietic stem cell transplantation (82%) or high-grade immunosuppressive treatment (18%) fulfilled the inclusion criteria. Upon rHZV vaccination, 88% (29/33) individuals mounted a significant antibody response exceeding the assay’s cut-off level for seropositivity (*p* < 0.0001). We detected higher geometric mean antibody concentrations after three compared to two doses. However, 12% remained below the assay’s cut-off level and were therefore considered non-responsive. **Conclusions**: The rHZV is immunogenic in VZV-seronegative immunocompromised individuals and therefore presents a valid option to induce seroconversion. However, antibody testing in high-risk groups should be considered to identify humoral non- and low responders.

## 1. Introduction

Persons with primary or secondary immunodeficiency are at higher risk of severe and complicated primary infection with varicella zoster virus (VZV) as well as VZV reactivation [1,2,3,4,5]. Among immunocompromised patients, especially those with transplants, malignancies or targeted therapies such as Janus kinase inhibitors have an up to 9-fold higher incidence of herpes zoster and postherpetic neuralgia relative to the total population [4,5,6,7,8,9,10]. Particular attention must be paid to those immunocompromised patients who lack an adequate VZV-specific immunity, indicated by VZV seronegativity (i.e., antibodies below the assay’s cut-off/detection level), and are therefore at risk of severe VZV infection/disease. Seronegativity can originate from either lack of previous VZV infection or vaccination, primary vaccination failure, or early antibody waning after infection or vaccination. Furthermore, a loss of antibodies and pre-formed long-lasting plasma cells as well as memory cells might follow immunoablation preceding hematopoietic stem cell transplantation (HSCT).

However, the use of the live-attenuated VZV vaccine to prevent varicella infection is contraindicated for severely immunocompromised persons [7,11,12,13,14], thus leaving VZV-seronegative immunocompromised patients without an option to vaccinate against primary varicella infection and thus at increased risk.

With respect to endogenous varicella reactivation, i.e., herpes zoster, a recombinant subunit vaccine (rHZV-Shingrix^®^) has been developed to prevent herpes zoster and postherpetic neuralgia. This vaccine replaced the former live-attenuated virus vaccine (Zostavax^®^) used for herpes zoster prevention and is safe to use in immunocompromised patients [6,15]. The rHZV targeting the VZV glycoprotein E antigen (gE) contains the adjuvant AS01B to enhance immunogenicity and is licensed for adults at increased risk for VZV reactivation, due to aging (≥50 years) or immunosuppression (≥18 years), in a two-dose schedule [15,16]. This vaccine demonstrated an overall vaccine efficacy of 97.2% in preventing herpes zoster in individuals above the age of 50 years [17]. In the elderly (≥70 years), vaccine efficacy was still 91.3% against zoster and 88.8% against postherpetic neuralgia [18]. Even in immunocompromised individuals (≥18 years), such as patients after autologous HSCT or with hematologic malignancies, a robust immune response [7,19,20,21,22] with high vaccine efficacy against zoster and postherpetic neuralgia (68% and 87%, respectively) was shown [23,24].

The rHZV is not licensed to prevent VZV primary infection, but, as antibody determination is generally not required prior to vaccination, the vaccine might, in rare cases, also be administered to people without previous contact to VZV [16]. Data on the immunogenicity after rHZV vaccination in initial VZV-seronegative immunocompromised patients are scarce and currently only available for solid-organ transplant patients [5]. In these cases, the use of rHZV remains the only safe option for vaccination, albeit as off-label application. However, according to experts’ opinion, more evidence and data on the protection against varicella infection are required to support the recommendation to use rHZV for primary vaccination in immunocompromised patients [16].

In the current study, we aimed to retrospectively evaluate antibody responses in immunocompromised individuals who previously tested anti-VZV-IgG negative and received rHZV vaccination in our outpatient vaccination clinic for patients with medical risk conditions. Immunogenicity data were analyzed from a cohort of patients with underlying hemato-oncological malignancies or autoimmune diseases, with their respective treatments, following two or three doses of rHZV. However, clinical outcome on the prevention of infection requires a separate evaluation.

## 2. Materials and Methods

### 2.1. Study Population and Study Design

For this retrospective explorative cohort study, we first screened the database from our outpatient clinic at the Institute for Specific Prophylaxis and Tropical Medicine (ISPTM) for all patients aged 18 years and above, who received two or three doses of rHZV (Shingrix^®^; GSK, London, UK) intramuscularly (in the deltoid muscle) between 1 March 2018 and 8 January 2024 (*n* = 952). Only immunocompromised individuals who tested seronegative for VZV-IgG antibodies (below the assay’s cut-off) before their first rHZV dose and with a VZV IgG test result at least four weeks after their last dose of rHZV were eligible, including one subject just before starting on immunosuppressive treatment. We excluded patients with intravenous human immunoglobulin substitution (IVIG) or varicella zoster immunoglobulin (VZIG) therapy in their history and one person who had received a solid organ transplantation. For the analysis, 33 patients were available (Figure 1). We assessed safety by retrospectively extracting documented adverse events from patients’ charts.

This study was approved by the ethics committee of the Medical University of Vienna on 27 April 2023 with an amendment for the analysis period on 12 January 2024 (Ethics Nr.: 1255/2023) and was conducted in accordance with ICH and GCP guidelines and with the applicable local regulatory requirements.

### 2.2. Vaccination Schedule

Two doses of rHZV were applied i.m. with an interval of at least one month, according to licensure. Only in HSCT patients, we initially offered three doses of rHZV at least one month apart, following a publication on the immunogenicity and efficacy of three rHZV doses after autologous HSCT [25]. Upon further published data in HSCT patients, a two-dose schedule has been implemented as routine schedule at our outpatient clinic. Patients with other diagnoses received the licensed two-dose regimen on a routine basis.

### 2.3. Data Extraction

The retrospective analysis of the antibody levels before the first and after the second or third rHZV vaccine dose in initially VZV seronegative patients following rHZV was performed by extracting rHZV vaccination history and VZV serology results from the database of the outpatient clinic of the ISPTM. Additionally, basic demographic data, including age, gender, underlying clinical conditions, medical and vaccination history and immunosuppressive medication, were extracted from patients’ charts (Table 1).

### 2.4. Analysis of Specific VZV IgG Levels

Anti-VZV-IgG antibody levels before the first and at least four weeks after the last (second or third) dose of rHZV were quantified according to routine diagnostic testing at the Center of Virology, Medical University of Vienna, Austria. Antibodies were measured using a CE-IVD certified, commercially available ELISA with VZV-lysate pre-coated plates (Euroimmun Medizinische Labordiagnostik, Lübeck, Germany), according to the manufacturer recommendations. As specified by the manufacturer, values < 140 IU/L were assessed as negative, >200 IU/L as positive and the values in between as borderline. In eight cases, VZV IgG levels were derived from an external laboratory, indicating values with respective interpretation according to the assay system used (*n* = 5 negative according to the test cut-off, *n* = 3 positive).

### 2.5. Endpoints

The primary endpoint was the quantitative result of VZV IgG ELISA more than four weeks after the last dose (second or third) rHZV in immunocompromised patients who tested negative before vaccination. The secondary endpoint was the percentage of seropositive participants, defined by a positive IgG ELISA result after vaccination.

A key question of this study was whether an antibody increase to a positive result, i.e., >200 IU/mL, can be established by vaccination with two or three doses of rHZV in these high-grade immunosuppressed patients in a routine clinical setting.

### 2.6. Statistical Analysis

The statistical analysis was performed using Stata 17.0 and figures were produced using GraphPad Prism 9.3.1. For calculation of the geometric mean concentration (GMC), a generalized linear model (GLM) was used with number of vaccinations as factor and time since last vaccination, age and sex as covariates. Wald 95% confidence intervals (CI) of the GMCs were computed, adjusted for the covariates. In five cases, anti-VZV IgG concentrations below the assay’s cut-off level were reported from another lab. If no values were specified in these cases, an arbitrary value of 70 IU/L (half of the cut-off) was used. For analysis of fold increase, the log pre-vaccination concentrations were used as offset; otherwise, the analysis was equivalent to that specified above.

All additional variables, including the increase from negative to positive values, were descriptively presented in total and as absolute and relative frequencies for nominal data and as median plus interquartile ranges (IQR) for metric data.

## 3. Results

### 3.1. Study Population and Demographic Data

Between 1 March 2018 and 8 January 2024, 952 individuals received rHZV vaccinations at our outpatient clinic that focused on immunocompromised patients. Of those, 91 initially tested seronegative for anti-VZV IgG (below the cut-off of the applied assay) before their first dose of rHZV, and were classified as at risk patients. Of these, 41 patients were subsequently tested for VZV IgG at least four weeks after receiving their last (second or third) dose of rHZV. A further eight patients did not meet the inclusion criteria and were excluded. In total, 33 patients were included in the final analysis (Figure 1).

The median age at the first dose of rHZV was 53.0 (43.0–61.0 IQR) years and 51.5% of the study group were female. The majority of patients (81.8%, *n* = 27) had a chronic hematological disease and had received HSCT. Most HSCT recipients (64.7%, *n* = 22) received three doses, starting three months after HSCT, on a routine basis following an early publication on HSCT patients (Table 1). For all patients, the median interval between last dose of rHZV and blood draw for VZV IgG testing was 9.6 months (5.1–13.9 IQR). No severe adverse events following immunization were recorded in the patient charts.

### 3.2. Humoral Immune Response

With respect to an antibody increase exceeding the assay’s cut-off level, 29 of the 33 individuals who previously tested negative (below 140 IU/L) achieved a positive VZV-specific IgG result (above 200 IU/L) at least four weeks after their last rHZV dose, resulting in a seropositivity rate of 88% (95% CI: 72–97%) (Figure 2A). However, four patients (12%) remained seronegative after two (*n* = 2) or three (*n* = 2) doses of rHZV (Table 2).

The GMC for all patients increased from 52 IU/L (95% CI: 40–68 IU/L; Figure 2B) before vaccination to 1445 IU/L (95% CI: 969–2157 IU/L; Figure 2B) after the last vaccination (corrected for distance from vaccination). Patients receiving two doses had significantly (*p* = 0.032) lower GMC (751 IU/L 95% CI: 354–1550 IU/L vs. 2001 IU/L 95% CI: 1201–3333 IU/L) than those who received three doses (Figure 2C). The antibody increase was highly significant (*p* < 0.0001), with a 28-fold GMC increase (95% CI: 17–43-fold) (Figure 2B). Yet, the antibody increase did not differ significantly (*p* = 0.533) between those receiving two doses (20-fold 95% CI: 9–46-fold) and three doses (32-fold 95% CI: 18–58-fold).

The interval between the IgG level assessment and the last vaccination had a significant impact on the IgG level (*p* = 0.001). Per month increase of the interval, antibody levels decreased by 10.9% (95% CI: 5.4–16.2%). Non-responders (*n* = 4) had a median (IQR) of 20.9 (12.1–28.8) months between their last dose of rHZV and the time point of VZV IgG testing, while for responders (*n* = 29) it was 9.1 months (4.1–11.7 months; *p* = 0.010). However, after correction for time from the last vaccination, the non-responders had expected levels of VZV IgG below 200 IU/L at one month after vaccination.

Patients with HSCT tended to have higher increases of IgG concentrations than the other patients (29-fold, 95% CI: 17–50-fold vs. 21-fold, 95% CI: 7–65-fold), although this was not statistically significant (*p* = 0.069). Since it takes several months to years for the immune system to reestablish after HSCT, we analyzed the median (IQR) interval between HSCT (*n* = 27) and the application of the first dose of rHZV; it was 10.0 (8.0–18.0) months for all patients, 7 (6 and 8 months) for non-responders (*n* = 2) and 11.0 (8.0–18.5) months for responders (*n* = 25), respectively. Although non-responders received their first dose somewhat earlier with respect to HSCT than responders, this difference was not significant. Despite this difference, the interval between HSCT and first rHZV dose had no significant effect on increase of IgG concentrations (*p* = 0.540) overall. In addition, allogeneic versus autologous HSCT showed no difference of the immune responses (*p* = 0.321).

## 4. Discussion

Encountering VZV-seronegative immunocompromised patients in our outpatient clinic, we aimed to retrospectively analyze their antibody responses to rHZV. Reasons for their VZV-seronegativity in our cohort remain unidentified, but could have been due to the absence of previous VZV contact or vaccination (naïve persons), primary vaccination failure, or the loss of antibodies after infection or vaccination due to early antibody waning in relation to the immunosuppressive therapy. Although we cannot investigate whether or which participants are truly naïve to VZV, they apparently are at increased risk for infection and/or severe disease.

At present, rHZV is not licensed for primary vaccination against VZV in seronegative individuals. Due to the contraindication of live-attenuated varicella vaccination in immunocompromised patients, off-label use of rHZV remains the only option to induce antibody responses against VZV in this special patient cohort [5,16]. However, the protection capacity of the induced immune responses directed against the rHZV vaccine antigen glycoprotein E (gE) against primary varicella infection is, as yet, unknown. So far, data on the antibody response following rHZV vaccination in VZV IgG negative persons under immunosuppression are limited to solid organ transplant recipients [5]. Therefore, the aim of this study was to assess antibody responses and seropositivity rates following rHZV in patients with immunodeficiency or with immunosuppressive or immunomodulatory therapies who previously tested negative for VZV IgG.

In our retrospective cohort study, the majority of participants (88%) mounted a significant increase in VZV IgG levels and reached the threshold of seropositivity of the assay. Our results are supported by a study in 23 anti-VZV-gE-seronegative solid organ transplant (SOT) recipients who showed a significant increase in antibody immune response, with a positive seroresponse in 55% four weeks after the administration of two doses of rHZV [5]. To our knowledge, this is the only trial on immune responses to rHZV in gE-seronegative immunocompromised individuals so far. Although our study now provides more data on this topic, as a limitation, we assessed VZV-specific antibodies with an ELISA using lysate and not gE as the target antigen. However, an earlier report demonstrated a high correlation between a gE antigen-specific ELISA and an ELISA using VZV-lysate as target antigen to measure the VZV-specific humoral immune response [26].

Due to the already published data, we excluded one person who had received a solid organ transplantation from our analysis. Results from a phase 1/2 study described one individual post autologous HSCT who was initially seronegative against gE and seroconverted after the first dose of an rHZV adjuvanted with AS01E (containing half the amount of the adjuvant included in the now licensed vaccine containing AS01B) [27]. In a subsequent phase 3 trial, pre-vaccination seropositivity rates for gE are indicated in adult autologous HSCT recipients, but seroconversion rates for baseline negatives are not mentioned [7]. However, peak antibody levels one month after the last dose of rHZV were reported, which subsequently declined.

We here demonstrate that the interval between IgG level assessment and the last vaccination had a significant impact on the IgG level, with a decrease by 10.9% per month increase of the distance. The patients we report as non-responders (*n* = 4) had longer intervals (median 20.9 months) between their last dose of rHZV and the blood draw for VZV IgG testing than responders (*n* = 29; 9.1 months), suggesting that fast antibody waning could have led to the seronegative results. However, further analysis following correction for interval-length from last vaccination rather points towards non-responsiveness (either intrinsic or extrinsic) and not early/fast antibody waning, as antibody concentrations were equal or even lower than pre-vaccination values in these patients.

Our study further showed that three vaccine doses led to higher antibody levels than two doses in highly immunocompromised patients. However, patient characteristics differed substantially, as three doses were applied only to HSCT patients (according to Winston et al.) [25]. The practice changed at our outpatient department when more data on the immunogenicity and efficacy of a two-dose schedule for these patients became available [7,23]. All other patients received a two-dose schedule. Therefore, we cannot draw conclusions in relation to which schedule may be superior with respect to quantity of antibody responses. Nevertheless, for the optimal management of non-responders to rHZV, these patients will require monitoring for symptoms of VZV infection or reactivation, to ensure prompt treatment. In addition, it will be crucial to clarify whether they might benefit from additional antiviral prophylaxis and/or an additional rHZV dose.

Importantly, we detected no safety concerns following the off-label use of two or three doses of rHZV in VZV seronegative individuals, and therefore confirm previously reported data for SOT patients and patients after autologous HSCT [5,7,23,25].

Limitations of our study are its retrospective character, the possibly incomplete information on prior VZV vaccination, infection or reactivation, the rather small sample size that does not allow for detailed subgroup analysis, and the use of a lysate rather than a gE-specific ELISA. A bias for the selection of included participants may arise from the fact that our clinic specializes in immunocompromised patients that are usually referred. Indeed, we used a test that was not optimized for the rHZV vaccine antigen gE, but the at least 5-fold (and up to 300-fold) increase in antibody levels in a lysate ELISA after rHZV indicates a strong humoral response against gE. Since assays to determine the cellular response to rHZV are not available on a routine basis, we were not able to present cellular analyses in our retrospective study. Thus we cannot exclude the possibility that our patients who did not reach antibody levels above the cut-off may nevertheless have experienced a cellular response, as described in the literature [28]. Accordingly, data by L’Huillier et al. from SOT patients without a gE-specific seroresponse demonstrated a significant increase of VZV-specific cellular responses following rHZV [5].

Our study did not evaluate the clinical effectiveness of rHZV to prevent varicella infection or zoster reactivation in immunosuppressed individuals, but quantified the humoral immune responsiveness in patients with absent VZV-specific humoral immunity. Therefore, we cannot predict if immunocompromised seronegative patients vaccinated with rHZV would benefit from a booster with the live-attenuated VZV vaccine upon re-established immunocompetence. In immunocompromised patients with previous VZV exposure, vaccine responders can be assumed to be protected against VZV reactivation, based on data from patients with hematological malignancies, and even after autologous HSCT [7,23,25]. However, the long-term protection in the immunocompromised is still unclear, and so far, data are only available for up to 21 months in highly immunocompromised patients [23].

## 5. Conclusions

Taken together, our results suggest the use of rHZV in VZV seronegative immunocompromised patients as a safe option to induce immune responsiveness to VZV. The data further suggest antibody testing in high risk groups to identify non- or low-responders, who may require further vaccine doses or prophylactic antiviral treatment.

Further studies on the longevity of the humoral and cellular immune response in addition to the clinical efficacy are urgently needed to provide evidence-based recommendations for patients with contraindication to live-attenuated VZV vaccination. Ideally, data can be generated from patient groups with different underlying diagnoses and immunocompromising treatments.

## Figures and Tables

**Figure 1 vaccines-13-00737-f001:**
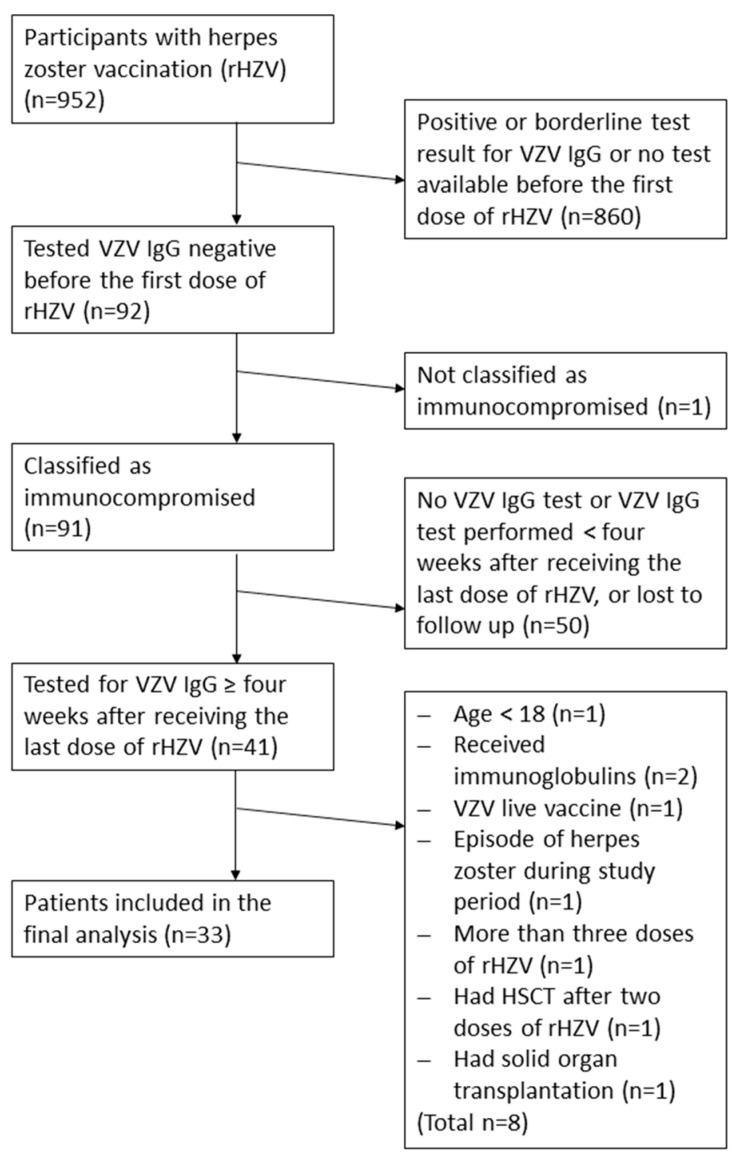
Flowchart of participants’ selection. IgG, immunoglobulin G; rHZV, recombinant herpes zoster vaccine; VZV, varicella zoster virus; HSCT, hematopoietic stem cell transplantation.

**Figure 2 vaccines-13-00737-f002:**
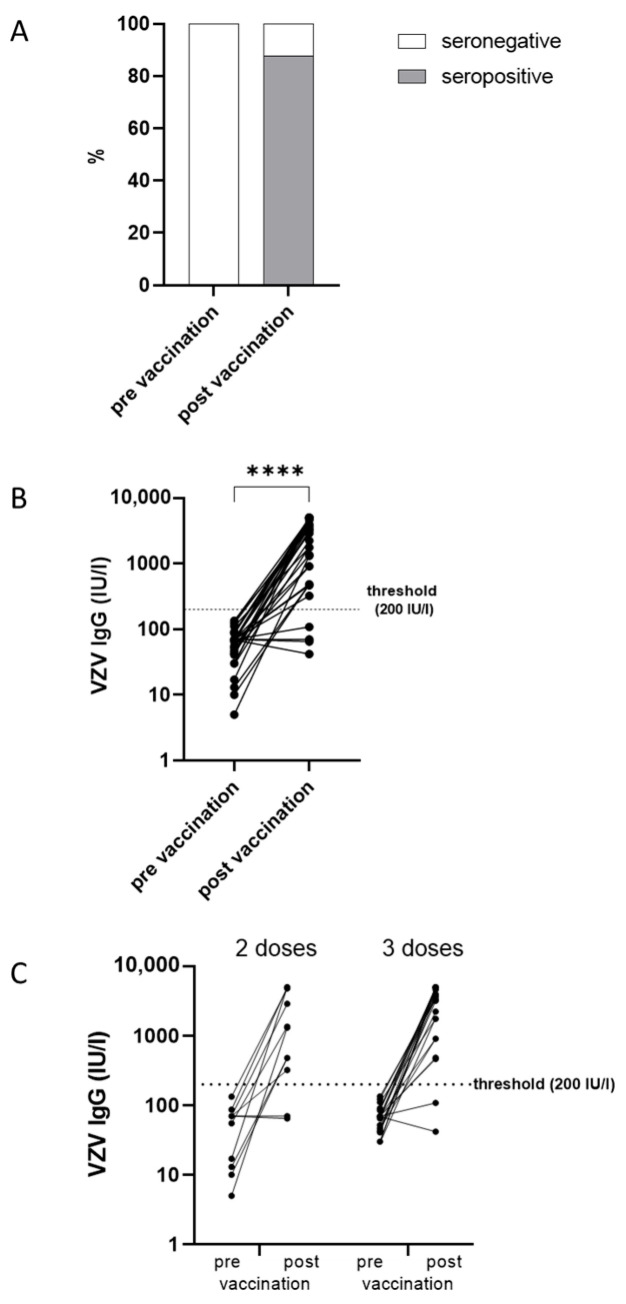
VZV-specific antibody responses. (**A**). VZV seropositivity rate prior and post vaccination with rHZV in % (*n* = 33). (**B**). Specific VZV IgG levels in IU/L prior and post vaccination with rHZV (*n* = 33). (**C**). Specific VZV IgG levels in IU/L prior and post vaccination with rHZV after two (*n* = 11) or three doses of rHZV (*n* = 22). The threshold for seropositivity is 200 IU/L. (**** *p* < 0.0001). rHZV, recombinant herpes zoster vaccine; VZV, varicella zoster virus; IgG, immunoglobulin G.

**Table 1 vaccines-13-00737-t001:** Patient characteristics.

Demographics		
Patients, *n*, %	33	100.0
Age at first dose of rHZV in years, median (IQR)	53.0 (43.0–61.0)	-
Female, *n*, %	17	51.5
Male, *n*, %	16	48.5
Interval between last dose of rHZV and VZV IgG testing in months, median (IQR)	9.6 (5.1–13.9)	-
Main diagnosis	*n*	%
Hematological disease	27	81.8
MM (multiple myeloma)	9	27.3
ALL (acute lymphoblastic leukemia)	1	3.0
AML (acute myeloid leukemia)	9	27.3
NHL (non-Hodgkin lymphoma)	1	3.0
DLBCL (diffuse large B-cell lymphoma)	1	3.0
MDS (myelodysplastic syndrome)	3	9.1
Osteomyelofibrosis	2	6.1
Plasma cell leukemia	1	3.0
Collagenosis	1	3.0
Multiple sclerosis	3	9.1
Rheumatoid arthritis	1	3.0
Uveitis	1	3.0
Therapy		
HSCT (hematopoietic stem cell transplantation) ^Δ^	27	81.8
Allogenic	17	51.5
Autologous	10	30.3
HSCT with maintenance therapy *	13	39.4
HSCT without maintenance therapy	14	42.4
Immunosuppression other than HSCT	6	18.2
One immunosuppressive drug **	2	6.1
More than 1 immunosuppressive drug ***	3	9.1
Immunosuppressive medication planned ****	1	3.0
Number of Shingrix^®^ doses		
Two doses of rHZV	11	33.3
Three doses of rHZV	22	66.7
Vaccination history before initial VZV IgG testing		
VZV live vaccine (2 doses) ^□^, patients *n*, %	1	3.0
rHZV (1 dose) ^○^, patients *n*, %	1	3.0

^Δ^ all patients with hematological diseases had a HSCT. * thalidomide analogues (pomalidomide/lenalidomide) *n* = 7; tyrosine kinase inhibitors (imatinib) *n* = 1; Janus kinase inhibitors (ruxolitinib) *n* = 2; glucocorticoids (prednisolone/dexamethasone) *n* = 5; calcineurin inhibitors (ciclosporin) *n* = 2, proteasome inhibitors (carfilizomib) *n* = 1; anti-CD38 (daratumumab) *n* = 1; B-cell maturation antigen (BCMA) antibody (belantamab mafodotin) *n* = 1. ** dimethyl fumarate *n* = 1; pyrimidine synthesis inhibitors (teriflunomide) *n* = 1. *** anti-CD20 (rituximab) *n* = 1—started one month after last rHZV dose; purine metabolism inhibitors (methotrexate) *n* = 2; IL-6 receptor antibodies (tocilizumab) *n* = 1; glucocorticoids (prednislolone) *n* = 3; anti-CD80/CD86 (abatacept) *n* = 1—ended two weeks before first rHZV dose; Janus kinase inhibitors (tofacitinib) *n* = 1. **** dimethyl fumarate. □ administered 12 months before initial negative VZV IgG testing. ○ administered 48 months before initial negative VZV IgG testing and before autologous HSCT.

**Table 2 vaccines-13-00737-t002:** Non-responder characteristics.

Characteristic	*n* = 4
Gender	Female (*n* = 4)
Age at first dose of rHZV in years, median (IQR)	54.0 (35.5–67.3)
Number of doses of rHZV	2 (*n* = 2), 3 (*n* = 2)
Diagnosis	Acute myeloic leukemia, collagenosis, rheumatoid arthritis, multiple myeloma
Interval between last dose of rHZV and VZV IgG testing in months, median (IQR)	20.9 (12.1–28.8)
Therapy ± immunosuppressive medication	Allogenic HSCT plus lenalidomide (*n* = 1) Autologous HSCT (*n* = 1) Methotrexate plus tofacitinib plus prednisolone (abatacept—until two weeks before first dose of Shingrix^®^) (*n* = 1) Rituximab (starting one month after second dose of Shingrix^®^) plus prednisolone 50 mg (*n* = 1)

## Data Availability

Data are made available conditional on ethics approval.

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
