# Peer review of "Antibody Responses Following Primary Immunization with the Recombinant Herpes Zoster Vaccine (Shingrix®) in VZV Seronegative Immunocompromised Adults"

_vaccines, 2025, doi:10.3390/vaccines13070737_

Round 1
Reviewer 1 Report
Comments and Suggestions for Authors
The authors should be congratulated for their innovative research reported in their concisely written manuscript.
Qusetions & Comments
Page 4, line 109. To my knowledge the licensed interval for the second dose is 2 to 6 month from the first dose. Authors should verify their statement.
Page 4, line 115. Authors should clarify if the history of prior VZV Infection (shingels or HZ) and receipt of childhood vaccination against VZV was elucidated before vaccination.
Author Response
We thank the reviewers for their constructive remarks and their accurate check of our manuscript. We addressed all the comments raised by the reviewers to our manuscript “Antibody responses following primary immunization with the recombinant herpes zoster vaccine (Shingrix®) in VZV sero-negative immunocompromised adults” in the following point-by-point reply. Changes made are highlighted in the revised version of the manuscript.
Point-by-Point reply
Reviewer 1:
Comment 1: Page 4, line 109. To my knowledge the licensed interval for the second dose is 2 to 6 month from the first dose. Authors should verify their statement.
Response 1: Yes, indeed the licensed interval for the second dose is two to six months. However, according to the product information of Shingrix (EPAR) the interval between the two doses may be one to two months for patients who are or might become immunodeficient or immunosuppressed due to disease or therapy.
Comment 2: Page 4, line 115. Authors should clarify if the history of prior VZV Infection (shingels or HZ) and receipt of childhood vaccination against VZV was elucidated before vaccination.
Response 2: Patient’s charts were checked for previous VZV vaccination and infection, 2 patients had a documented VZV vaccination (now listed in Table 1) but no shingles episode was reported. However, patients came to our outpatient clinic when they were adults and just before or already after immunosuppressive therapy commenced. Therefore, it cannot be excluded that a patient underwent shingles during childhood. Shingles infection was documented for one eligible patient – and this patient was excluded from the analysis (Figure 1). We now stated among the limitations, that we cannot guarentee completeness of infection or vaccination history.
Reviewer 2:
Major comments:
Comment 1: Please discuss the relevance of correlation between total antibody concentration and neutralizing antibody titers. Please describe why the study did not address the quantitation of VZV neutralizing antibodies post immunization of the immunocompromised cohort.
Response 1:
We agree with the Reviewer that measuring neutralizing antibody titers is considered the gold standard in antibody assessments (in terms of measuring the antibodies´ functionality and using an assay with the highest specificity). However, our study aimed to quantify the humoral immune response to the herpes zoster vaccine (Shingrix) by measuring antibodies as a correlate of this response, rather than quantifying their functionality as a correlate of protection. The study employed a retrospective design utilizing antibody measurements from routine diagnostics and neutralization assays are not part of these routine procedures.
Minor comments:
Comment 2: Line 63; add space between “years” and “[17]
Response 2: Done
Comment 3: Line 64; remove space between “post” and “herpetic” to become one word as written throughout the manuscript.
Response 3: Done
Comment 4: Line 67; remove hyphen from “post-herpetic”
Response 4: Done
Comment 5: Line 98 – 99; authors state that “ for the analysis, 33 patients were available”. Meanwhile, in Figure 1 last text box, it is listed that “Patients included in the final analysis (n=34)”. Please align text and Figure 1 the final patient number.
Response 5: Sorry for this error that was due to inclusion of an outdated version.
Comment 6: Line 112; correct HSCT.[25]
Response 6: HSCT.[25] has been corrected to „HSCT [25].“ In Line 112.
Comment 7: Line 116 – 121; adjust font type and size to align with font and size used throughout the manuscript
Response 7: Done
Comment 8: Line 173 and Figure 1; Authors indicated that 8 patients did not meet inclusion criteria while in Figure 1 stated only 7 patients did not meet inclusion criteria. Align between text and figure the final number of patients being excluded or add the exclusion criterion for the 8th patient to Figure 1.
Response 8: Sorry! Again due to using an older version of the figure.
General comment:
1) Please add the date the ethical approval was obtained to the manuscript.
We now added the date of the initial approval (Jan. 27th, 2023) and for the amendment of the analysis period on Jan 12th, 2024 (line 102).
Reviewer 2 Report
Comments and Suggestions for Authors
Major comments:
Please discuss the relevance of correlation between total antibody concentration and neutralizing antibody titers. Please describe why the study did not address the quantitation of VZV neutralizing antibodies post immunization of the immunocompromised cohort.
Minor comments:
Line 63; add space between “years” and “[17]
Line 64; remove space between “post” and “herpetic” to become one word as written throughout the manuscript.
Line 67; remove hyphen from “post-herpetic”
Line 98 – 99; authors state that “ for the analysis, 33 patients were available”. Meanwhile, in Figure 1 last text box, it is listed that “Patients included in the final analysis (n=34)”. Please align text and Figure 1 the final patient number.
Line 112; correct HSCT.[25]
Line 116 – 121; adjust font type and size to align with font and size used throughout the manuscript
Line 173 and Figure 1; Authors indicated that 8 patients did not meet inclusion criteria while in Figure 1 stated only 7 patients did not meet inclusion criteria. Align between text and figure the final number of patients being excluded or add the exclusion criterion for the 8th patient to Figure 1.
Author Response

(The authors gave the same response as above.)
